# Connecting RNA-Modifying Similarities of TDP-43, FUS, and SOD1 with MicroRNA Dysregulation Amidst A Renewed Network Perspective of Amyotrophic Lateral Sclerosis Proteinopathy

**DOI:** 10.3390/ijms21103464

**Published:** 2020-05-14

**Authors:** Jade Pham, Matt Keon, Samuel Brennan, Nitin Saksena

**Affiliations:** 1Faculty of Medicine, The University of New South Wales, Kensington, Sydney, NSW 2033, Australia; jade.pham@student.unsw.edu.au; 2Iggy Get Out, Neurodegenerative Disease Section, Darlinghurst, Sydney, NSW 2010, Australia; mattk@iggygetout.com (M.K.); sam@iggygetout.com (S.B.)

**Keywords:** microRNA, amyotrophic lateral sclerosis, RNA metabolism, TDP-43, FUS, SOD1, RNA-binding proteins

## Abstract

Beyond traditional approaches in understanding amyotrophic lateral sclerosis (ALS), multiple recent studies in RNA-binding proteins (RBPs)—including transactive response DNA-binding protein (TDP-43) and fused in sarcoma (FUS)—have instigated an interest in their function and prion-like properties. Given their prominence as hallmarks of a highly heterogeneous disease, this prompts a re-examination of the specific functional interrelationships between these proteins, especially as pathological SOD1—a non-RBP commonly associated with familial ALS (fALS)—exhibits similar properties to these RBPs including potential RNA-regulatory capabilities. Moreover, the cytoplasmic mislocalization, aggregation, and co-aggregation of TDP-43, FUS, and SOD1 can be identified as proteinopathies akin to other neurodegenerative diseases (NDs), eliciting strong ties to disrupted RNA splicing, transport, and stability. In recent years, microRNAs (miRNAs) have also been increasingly implicated in the disease, and are of greater significance as they are the master regulators of RNA metabolism in disease pathology. However, little is known about the role of these proteins and how they are regulated by miRNA, which would provide mechanistic insights into ALS pathogenesis. This review seeks to discuss current developments across TDP-43, FUS, and SOD1 to build a detailed snapshot of the network pathophysiology underlying ALS while aiming to highlight possible novel therapeutic targets to guide future research.

## 1. Introduction

Amyotrophic lateral sclerosis (ALS) is a fatal, incurable, and highly heterogeneous neurodegenerative disease that deteriorates both upper and lower motor neurons (MNs), leading to extensive muscle weakness, atrophy, and paralysis [1]. More than 90% of all ALS cases are sporadic (sALS), with individuals often succumbing to respiratory failure 3–5 years after symptom onset; the remaining 10% represent familial cases (fALS). Unlike fALS, sALS has no delineated central mechanism, although multiple key molecular mechanisms have been uncovered including impaired proteostasis, nucleocytoplasmic transport defects, impaired DNA repair, vesicle-transport defects, neuroinflammation, and mitochondrial dysfunction [2]. However, we must highlight the significance of altered RNA metabolism in the causative chain of ALS development in neurodegeneration, as altered RNA metabolism proves to be both common and essential in precedence to key pathological events including later impairments in splicing, transport, and stability (Figure 1) [3].

### Examination of Amyotrophic Lateral Sclerosis (ALS) with a Focus on Proteostasis and microRNAs

Over the years, the number of neurodegenerative diseases that have been associated with pathological protein aggregates has increased. In ALS, more than 50 genes [4] have been linked to pathogenesis, with the most significant being superoxide dismutase-1 (SOD1), fused in sarcoma/translocated in liposarcoma (FUS/TLS), and transactive response DNA-binding protein (TDP-43), all of which share several properties related to RNA regulation, with the latter two true RNA-binding proteins themselves (RBPs). Several RBPs have been implicated in ALS, which may alone or in partnership play a role in disease progression [1,5]. These proteins form aggregates, which bear a close association with motor neuron loss in both familial and sporadic ALS, and has thus resulted in a protein-centric paradigm shift. This is supported by the presence of TDP-43 inclusions in 97% of ALS cases [6], which may contain other proteins (like those involved in stress response), although only a minority of sALS cases show either FUS- or SOD1-positive inclusion bodies [6,7,8,9], with the former eliciting no proven overlap with TDP-43 aggregates [10,11]. More interestingly, mutated TDP-43 only accounts for 3% of fALS and 1.5% of sALS [12], and therefore 95% of patients with positive TDP-43 inclusions fail to carry a mutation in TAR DNA-binding protein (TARDBP) [11,13]. Despite this almost ubiquitous presence of TDP-43 aggregates, TDP-43-centric sALS studies may be considered limited in the scope of a mechanistic understanding of ALS as biochemical alterations and correlations instead reflect a “multiRBP proteinopathy” [1,14], or perhaps to better generalize, a network, multi-proteinopathy that is rooted in RNA dysfunction. Moreover, the striking structural and functional similarities between RBP aggregates of TDP-43 and FUS/TLS (Figure 2 and Figure 3) reinforce the pivotal role that impaired and abnormal RNA metabolism plays in disease progression, despite the specific events preceding formation remaining poorly understood.

Interestingly, both SOD1 and several RNA-binding proteins including TDP-43 and FUS, have also been shown to contain aggregation-promoting prion-like domains that are able to rapidly associate. This property cements these RNA-modifiers as plausible instigators of proteinopathy. Recent studies have also highlighted the interference of pathological TDP-43 and FUS aggregates with normal stress granule formation upon mis-localization to neuronal cytoplasm [15,16], which amplifies the effects of aggregation as clearance is hindered. The proteostasis of RBPs appears to be one of the most significant occurrences in ALS-affected tissue, or of SOD1 in mutant SOD1 ALS, and is only supplemented by the numerous competing and integrated biological pathways that control miRNA biogenesis, protein folding, trafficking, and degradation present within and outside the cell.

Several miRNAs have also been put forth as potential diagnostic biomarkers for ALS in both the neuron and microglial cells including miR-155, Let-7a, and miR-125b [17,18]. Results from recent studies strengthen the impact that miRNAs might have on post-transcriptional modulation of genes linked to both inflammation and ALS [17,19]. As small, non-coding RNA molecules critical in regulating transcriptional and post-transcriptional mRNA expression [20], miRNAs exist as a second pertinent subfield under pathophysiological study. Following the strict regulation of biogenesis, intricate connections have been elucidated between miRNA and RNA-binding proteins, with essential regulatory complexes like Drosha in the nucleus and Dicer in the cytoplasm being shown to be involved with TDP-43. Drosha complexes with DGCR8 have also been associated with most TDP-43, which further suggests a more complex dynamic that miRNAs and protein pathologies share, especially in motor neurons (MNs) [21].

As we have described, the majority of ALS cases—over 97%, inclusive of both sporadic and familial [6]—include wild-type TDP-43 aggregates, so it is also important to note that mutations in *SOD1* account for 20% of familial ALS and 5% of sporadic disease [22,23], although more recent studies have suggested possible overestimation, determining SOD1 mutations to be in < 1% of sALS patients [24]. Mutations in *TARDBP* and *FUS* account for only 5–10% and 5% of fALS, respectively, varying among ethnicities, all of which cement the intrinsic role of RBPs genes in ALS pathology. Overall, mutations in SOD1, TARDBP, and FUS occur in < 10% of cases in population-based studies, while mutations in other genes are even more uncommon [25].

Given the functional synergies and dependencies between these proteins, this article reviews the current concepts toward understanding the role of these three major proteins (TDP-43, FUS, and SOD-1) and their relationship with RNA metabolism and microRNA in ALS. As a key pathological event, this can therefore promote a more holistic understanding of the pathogenesis of ALS, even amongst the extensive heterogeneity of phenotypes, and thereof can provide plausible research avenues for future therapeutic targets. As a result, this review is framed around microRNA biogenesis and regulation; the significance of ALS-associated proteins, their interrelationships, and non-coding RNA molecules; and the overall RNA dysregulation that contributes to cellular and network dysfunction in ALS.

## 2. MicroRNA: Biogenesis, Regulation, and Protein-Related Dysfunction

To date, microRNAs primarily operate through the translational repression and/or decay of mRNA transcripts via complementary base-pairing [26]. As negative regulatory switches for a multitude of essential biological processes, alterations in miRNA expression are reflected in the pathogenesis of many human diseases including cancer and neurodegeneration [27,28,29]. Understanding the mechanisms that regulate individual miRNA and protein expression will help elucidate pathways involved in human disease, and identifying the interactions between miRNA and prion-like RBPs could further consolidate its application in ALS pathogenesis. 

### Biogenesis of miRNA and Gene Regulation

MiRNAs follow a complex biogenesis (Figure 1), with the majority involving regulatory complexes like Drosha, in the nucleus, and Dicer, in the cytoplasm, both of which have been shown to be involved with TDP-43. Only about 1% of conserved miRNAs are involved in non-canonical pathways (Dicer and/or Drosha- independent), with the remainder either low in abundance or poorly conserved [30]. To date, however, no non-canonical miRNAs have been associated with fALS or sALS.

#### Canonical MicroRNA Biogenesis

Most miRNAs are transcribed from intergenic regions, introns, and exons by RNA polymerase II. The initial RNA transcript is a RNA precursor called a primary miRNA (pri-miRNA) [31,32,33,34] (Figure 1), which ranges from 200 nucleotides to several thousand nucleotides in length, and is known to form a highly-structured stem loop [35,36]. The cellular RNase III enzyme ‘Drosha’ cleaves this stem loop with the help of cofactor DGCR8 in vertebrates and ‘Pasha’ in invertebrates, with a recent study also elucidating the critical role of Heme in efficient pri-miRNA processing alongside DGCR8 [37,38,39,40] (Figure 1). The cleavage produces an RNA hairpin intermediate around 70 nucleotides, known as precursor-miRNA or pre-miRNA, with a characteristic two nucleotide 3′ overhang [40]. 

Following pre-miRNA production, a heterodimer consisting of exportin 5 (EXP5) and the GTP-bound cofactor, Ras-related nuclear protein (RAN), assists nuclear export, after binding the two nt 3′ overhang of pre-miRNA [41,42] (Figure 1). In the cytoplasm, another cellular RNase III enzyme, Dicer, binds to the structured DNA with co-factor transactivation response RNA binding protein (TRBP) to perform a second cleavage. The end-product is a two nt 3′ overhang approximately 17–22 bp double stranded RNA (dsRNA). One strand of the dsRNA remains bound to Dicer to form the mature miRNA while the other RNA strand is typically degraded. The remaining strand is then integrated into a protein complex involving an Argonaute (AGO), forming the RNA-induced silencing complex (RISC) with the help of Dicer [42]. Any miRNA strands with central mismatches or absent AGO2 are subsequently unwound and degraded [30].

Mature miRNA bound to the active RISC then binds to the target sites at the 3′ untranslated region (UTR) of a target mRNA, leading to direct inhibition of translation or mRNA target degradation (Figure 1) [43,44]. Vertebrate miRNAs only require partial complementarity between miRNA and target strands to effect translational repression, although it is critical to have a high degree of complementary base pairing of miRNA nucleotides through their “seed sequence”, which is identified at nucleotides 2–8 [45,46].

Examining this heavily conserved biogenesis ultimately illustrates the extreme degree of regulation of an intricate, essential, multi-step process that emphasizes both the scale of the cumulative effects of miRNA control and the potential for missteps at multiple points along production.

## 3. TDP-43, FUS, and SOD1 Relations with miRNA in ALS

### 3.1. TDP-43 and Regulation of miRNA Biogenesis

TDP-43 is a nuclear protein that shuttles between nucleus and cytoplasm and like FUS, is structurally and functionally similar to the heterogeneous nuclear ribonucleoproteins (hnRNPs), which are involved in RNA processing. TDP-43′s association with over 6000 mRNA targets—almost 30% of the entire human transcriptome—suggests a significant capacity to cause regulatory chaos when this gene becomes dysfunctional [47]. Pathologically, TDP-43 is ubiquitinated, cleaved, hyper-phosphorylated, and cytoplasmically translocated in affected neurons and glial cells of ALS and frontotemporal lobular dementia (FTLD) patients [13,47,48,49], while inclusions were also recently reported in skeletal and cardiac tissue of sALS patients [50] and those with known C9ORF72 expansion [51]. In ALS, TDP-43 inclusions also include toxic fragments of the C-terminal region after protein truncation, although rarely observed in spinal cord tissue and may therefore result from regional heterogeneity in the central nervous system (Figure 1) [52,53,54].

This raises the question as to whether mis-localization and subsequent nuclear depletion leads to a reduced supply of the protein for its nuclear role, or if pathology results from toxic cytoplasmic aggregates. In the nucleus, the role of TDP-43 has been implicated in DNA repair [55], mRNA splicing regulation, miRNA biogenesis, and processing [48,54]. One recent study by di Carlo et al. [56] illustrated the role of TDP-43 in maintaining Drosha stability, whereas another study by Kawahara and Mieda-Sato [21] identified TDP-43, but not FUS, as a component of nuclear Drosha complexes that contain DGCR8, a cofactor indispensable for pri-miRNA processing. The Drosha-DGCR8 complex, known as the Microprocessor, is regulated in its protein stability, nuclear localization, and processing efficiency [30]. This group found that TDP-43 facilitated the binding of the Microprocessor to a subset of pri-miRNAs, which resulted in their efficient cleavage into pre-miRNAs.

TDP-43, therefore plays a clear and essential role as a nuclear protein; pathologically, TDP-43 has a less understood role, as hypothesized to be translocated to the cytoplasm [57,58,59,60,61] due to calpain-initiated cleavage at the proposed ‘prion-prone’ C-terminal [62], especially in TARDBP-associated ALS. External to miRNA production, TDP-43 is involved with the formation of ribonucleoprotein (RNP) granules for mRNA transportation [63] and mRNA translational repression via complex formation such as with the ribosomal receptor for activated C kinase 1 (RACK1), or via translation factor sequestration into stress granules [54], which will be elaborated on later in this review. However, Kawahara and Mieda-Sato elucidated the association of cytoplasmic TDP-43 with the Dicer complex that contained TRBP, which is also essential for AGO2 recruitment in miRNA biogenesis. This interaction facilitated Dicer processing of specific precursor miRNAs (pre-miRNAs), namely a subset of pre-miRNAs whose production in the nucleus is regulated by TDP-43 via direct binding to their terminal loops. The results thereby highlight two conclusions: (1) nuclear TDP-43 is a facilitator in the production of a subset of pre-miRNAs by both Drosha complex interaction and direct binding to relevant primary miRNAs (pri-miRNAs), and (2) cytoplasmic TDP-43 plays a role in promoting the processing of selected pre-miRNAs, interacting with the Dicer complex and/or having directly bound to terminal loops. Overall, these observations support a previously uncharacterized role for TDP-43 in post-transcriptional miRNA regulation in both the nucleus and cytoplasm, with further studies supporting TDP-43 involvement in miRNA biogenesis for neuronal outgrowth [21].

Although Kawahara and Mieda-Sato revealed the association that TDP-43 and Drosha share, they noted that the interaction was only permitted by certain RNA species, or a direct protein–protein interaction, which occurs through the C-terminal tail (of amino acids 316–401). Treatment with RNase VI, A, or both, elicited a dose-dependent response, highlighting RNA to be crucial in the typical functioning of TDP-43 and Drosha. However, even upon excess RNase treatment and a proven dispensability of the RNA-binding domain of TDP-43 in vitro, some association remained. The study thus concluded that the interaction between TDP-43 and Drosha exists as both RNA-dependent and -independent, although still indispensable for pre-miRNA processing. This raises the question of the role that Dicer complexes without TRBP play: are they simply “floaters” in the bigger picture of miRNA biogenesis, or do they have an unspecified, yet critical role? One possibility may include the existence of a feedback system, where these non-TRBP Dicer complexes assist, should there be a shortage of certain miRNAs. Overall, this study demonstrates the unique function of TDP-43 not only in the nucleus, but also in the cytoplasm, which may have immense relevance in the compartmental shuttling of TDP-43.

#### TDP-43 and Associated miRNAs: Beyond Biogenesis

As evidenced through multiple studies conducted on induced pluripotent stem cell (iPSC)-derived neurons, TDP-43 affects several miRNA expression levels upon downregulation, and it is thus unsurprising that the regulatory effects of TDP-43 extend beyond biogenesis and toward direct associations with produced miRNA such as that of miR-let-7b [54,64,65]. TDP-43 knockdown elicited a downregulation of miR-let-7b in vitro, and upregulation of miR-663, although the latter was a result of direct binding to its precursor, rather than directly to its mature sequence [64]. A recent study [29] on post-mortem spinal cords of sALS patients also reflected a global reduction in mature miRNA, with two exceptions: miR-155 and miR-142-5p. Upregulated miR-155 has been examined in both pre-symptomatic and symptomatic stages of disease in SOD1 mice models, and is known to regulate microglial responses and NF-kB-controlled responses in neuroinflammation as well as other immune responses in diverse neurodegenerative diseases [66]. In examining the location of an upstream fault in miRNA maturation and production, the team further examined two more miRNAs—miR-577 and miR-Let-7e—and their primary miRNA transcripts. The expression of the two pri-miRNAs proved to be unchanged, which suggests a likelihood for error during processing, rather than alterations to gene expression [29]. Whether the role of TDP-43 in this scenario could be attributed to or supplemented by other RBPs is unknown, with one needing to consider similar downstream effects due to pathological stress granule dynamics and the miRNAs involved in such [29,67].

Considering the lack of categorization of miRNAs, and the numerous, yet specific, miRNAs involved in TDP-43 pathology and beyond, perhaps one should consider, at least in the scope of ALS, the grouping of miRNA into “families” of those associated with TDP-43 and at what stage of miRNA biogenesis this effect occurs. More research is encouraged to make such specific miRNA connections more valuable, as they present a promising target group for novel therapies.

### 3.2. FUS-TLS: Functional and Pathological Comparisons to TDP-43 in ALS

Less prevalent amongst cases is the presence and discussion of the similar RBP, fused in sarcoma/translocated in liposarcoma (FUS/TLS). Following the discovery of TDP-43 and its mutants in ALS, FUS was identified to be associated in both familial and sporadic forms of ALS, which held considerable significance as both proteins share intense functional similarities as RNA binding proteins and display mis-localization and aggregation as main features [12,68,69,70]. As both nuclear proteins and hnRNPs, they are mainly involved in functions such as nucleo-cytoplasmic shuttling, RNA transcription, translation, splicing, transport for local translation, and stress granule formation. Under pathological conditions, naturally occurring mutations in FUS and TDP-43 guide the formation of cytoplasmic aggregates that are known in different NDs including ALS [12]. Post-translationally, these proteins, however, undergo differing processes: TDP-43 is ubiquitinated, phosphorylated, acetylated, sumoylated, and cleaved, while pathological FUS is phosphorylated [71], sumoylated [72], and methylated at its C-terminal arginine residues [14]. Recently, it has been shown that the loss of FUS in the nucleus can impair alternative splicing and/or transcription, whereas dysfunction of FUS in the cytoplasm where it aggregates, especially in the dendritic spines of neurons, can cause mRNA destabilization [73,74,75,76].

#### Repercussions of FUS Dysfunctions on miRNA Biogenesis

Although TDP-43 inclusions are absent in ALS patients with FUS mutations [10,77,78] and therefore, neurodegeneration driven by mutant FUS may thus be independent of TDP-43 mislocalization [12,68], both FUS and TDP-43 share multiple functions (Figure 2 and Figure 3). Thus, it is not surprising that FUS is known to localize together with TDP-43 at the nuclear Drosha complex (Figure 1) [40], nor that FUS contributes to the regulation of a specific subset of miRNAs [79]. Later studies have identified the direct binding of FUS to nascent pri-miRNAs to recruit Drosha to transcriptionally active sites for further pri-miRNA processing including miR-9, miR-125b, and miR-132 [14,79]. Interestingly, the similarities between FUS interactions with Drosha and neuronal pri-miRNAs also extend to other critical RBP genes including FUS, TAF15, ATXN2, MATR3, hnRNPA2/B, EWSR1, hnRNPA1, and TIA1, which further supports the almost universal relevance of RBPs, miRNAs, and RNA metabolic dysfunction across both sporadic and familial forms of ALS, despite observed heterogeneity [26,80]. This may suggest that dysregulation of RNA metabolism—coupled with cytoplasmic mis-localization of RBPs, dysfunction in stress granule dynamics of RBPs, and increased propensity of aggregate-forming mutant RBPs—is at the heart of ALS pathogenesis [5].

Furthermore, FUS has been shown to promote optimal miRNA-mediated gene silencing via direct binding to certain miRNA and mRNA targets including mature miR-200c, with mutant FUS forms impairing AGO2 protein function in the miRNA-induced silencing complex (miRISC) [81]. As a conserved mechanism and having AGO2 involved across miRNAs, it is undetermined whether this silencing regulation by FUS entails a global impact, or is limited to selective silencing of specific miRNA or mRNA interactions. FUS, as an RBP like TDP-43, also shares similarity in its response to stress and in excessive subnuclear paraspeckle formation [82], both of which are intertwined with miRNA regulation and biogenesis. This will be discussed in more detail later in the review (see Section 4.2.1).

However, beyond biogenesis, loss-of-function effects related to reduced miRNAs including miR-375 and its target genes, have also been elicited in vitro in human MNs upon induction of mutant FUS [83]. Reduced miR-375 has particularly been associated with p53-related apoptosis in spinal MNs across ALS and spinal muscular atrophy (SMA), with enrichment instead elicited in human embryonic stem cells, protecting DNA damage-induced apoptosis in MNs [84]. This study hence confirms the potential for indirect interference of mutant RBPs and other RNA-modifying proteins in affecting gene expression via RNA metabolic dysregulation. Interestingly, a recent paper simulated an inflammatory environment to both mouse and human neural progenitor-derived astrocytes with exhibited WT-FUS overexpression, where they displayed more sensitivity to IL1β compared to the controls. This resulted in promotional effects on neuronal cell death and pro-inflammatory microglia, which confirms the belief that non-cell autonomous mechanisms like protein dysfunction and RNA dysregulation can ultimately drive neurodegeneration [85].

### 3.3. SOD1 and miRNA: Aggregation, Dysfunction, and Other Pathological Similarities to RNA-Binding Proteins in ALS

Unlike TDP-43 and FUS, superoxide dismutase (SOD1) is not classified as an RBP, despite sharing key roles (Figure 2); however, to better complete the picture of miRNA metabolic dysfunction in MN-affecting diseases, we must also discuss SOD1 in ALS, especially in relation to miRNA production and regulation. SOD1—the very first ALS-associated gene to be identified [22]—is a copper and zinc-containing protein with 153 amino acids, which functions to detoxify superoxide radicals by their dis-mutation into oxygen and hydrogen peroxide [86,87]. Wild-type SOD1 (SOD1^WT^) is not commonly associated with aspects of neurodegeneration including in ALS, although upon post-translational modification such as iper-oxidation, even if noninheritable, studies suggest the possibility of its additional toxic role in sALS [87,88], which confers with the existing literature on the prion-like potential of SOD1, both wild and mutant [87,89,90,91]. Interestingly, a novel review [92] into the role of SOD1^WT^ in Parkinson’s disease (PD) has sparked interest in its toxic ‘gain-in-function’, as aggregates of SOD1^WT^ were discovered in postmortem idiopathic PD neural tissue [93,94]. It is known that metallated wild-type and mutant SOD1 can readily oligomerize under loss of bound metals [95,96], especially if combined with oxidative stress [97,98]; from here, SOD^WT^ can then become dysfunctional, misfolding, and aggregating [92]. Notably, the study therefore suggests a perspective of SOD1 where the significance of non-genetic factors in ND pathogenesis is likely underestimated, particularly as in SOD1-related fALS, patient survival times prove largely varying and age-related [99]. To enlighten our understanding between mutant and wild-type SOD1 in ALS, this may hence suggest that such wild-type ‘gain-in-function’ of SOD1 aggregates are not ALS-specific. Further investigations into similar posttranslational modifications of SOD1^WT^ in other NDs and controls are therefore required to confirm this hypothesis, and to identify whether toxicity from SOD^WT^ build-up is merely a common result of neurodegenerative-related stress.

Located in exon sequences, SOD1 mutations were initially thought to be purely due to a loss of physiological function, which would result in an increase in oxidative stress and eventual excitotoxicity. However, recent investigations have instead illuminated a gain of new toxic properties through its role in protein aggregation, as a result from its mutant-induced co-aggregation with other proteins [100] such as the RBPs, FUS, and TDP-43 and others with prion-like domains. More importantly, the mutations in SOD1 cause alteration in protein stability and their propensity to aggregate, which is similar to TDP-43 and FUS, while also correlating with ALS disease development; this hence implies the intrinsic functional integrity of these proteins needed in vivo, as any instability disrupting their integrity may not only have a bearing on each other, but also on the maintenance of protein homeostasis with the cell. Moreover, the ‘gain in function’ of mutant SOD1 highlights this similarity to RBPs with its consequential effects on RNA metabolism, despite lacking RNA-binding motifs. Thus, although not strictly an RBP, mutant SOD1 emerges with similar capabilities in triggering dysfunction across miRNA biogenesis and mRNA stabilization and support [101,102], especially considering noteworthy interactions between wildtype SOD1 and both TDP-43 and FUS, as later discussed.

Moreover, in the literature, it is generally accepted that SOD1 and TDP-43 inclusions are exclusive of one another, particularly when examining mutant SOD1 patients, as is more common with fALS [10]. However, recent studies [6,7,8,9,11] suggest that TDP-43 inclusions are also displayed in cell, mouse, and human models of mutant SOD1 ALS. Jeon et al.’s study [6] particularly proposes the potential for SOD1 mutations in altering TDP-43 metabolism after end-stage correlative observations of insoluble SOD1 fractions with TDP-43 C-terminal fragments, and through such, poses an avenue for indirect miRNA dysregulation by mutant SOD1. Here, even if there lies the possibility of SOD1 sequestering either protein, RNA, or both, by chance, further research should be conducted in identifying the consistencies amongst SOD1 aggregates, even if less commonly found in sALS patients; more importantly, this would confirm whether TDP-43, if not other RBPs, is a key point of convergence across ALS phenotypes, especially if specific treatment responses prove highly varying. This would be especially interesting as Yamashita et al. [62] had previously investigated ADAR2 knockout mice against SOD1 transgenic mice, where significant differences between calpain and calpastatin expression profiles suggested a mechanistic difference between SOD1-associated ALS and sALS. Future investigations should aim at clarifying this ‘point of highest convergence’ in heterogeneous ALS, whether this would involve TDP-43 regulation, or if further downstream or upstream, RNA or miRNA dysfunction may instead be worthwhile considerations for targeted therapy.

Interestingly, as miRNAs do regulate various genes involved in oxidative stress response, and as the converse also proves true, it is important to consider the strength of interrelationship between miRNA dysregulation and ALS pathogenesis, especially as oxidative stress is so strongly related to both SOD1 and other parts of pathogenesis, potentially leading to other processes of degeneration, despite often being examined in separation to RNA dysregulation [103]. The significance of connecting miRNA to the pathology of ALS therefore cannot be underestimated, even beyond SOD1 mutation, particularly when miRNA involved in mitochondrial/oxidative stress such as miR-338-3p [16,104] and miR-142-5p [105,106], and neuroinflammation, like miR-155 [107], have been present amongst both sALS and fALS pathology in both animal models and humans. The regulation of the Nrf2-ARE pathway is also of interest [103,108] as its link to ALS through redox-related gene regulation is regulated by several miRNAs including direct miRNAs like miR-27a and miR-34a, or indirectly by miR-7 and miR-494 [109,110,111,112].

#### Repercussions of SOD1 Dysfunction on miRNA Biogenesis and Regulation

In sALS individuals, SOD1 is not as commonly discussed as it is in fALS, where it mediates the disease in around 20% of affected individuals [113].

In 2009, a paper by Williams et al. highlighted how miRNA-206 delays ALS progression in G93A-SOD1 mice, and was significant in confirming the overall importance of miRNA involvement in ALS pathogenesis [114]. Mediated by muscle-derived factors and skeletal muscle-specific, miR-206 is key in the efficient compensatory regeneration of neuromuscular synapses post-injury, and dramatically upregulated miR-206 levels in ALS mouse models proved to coincide with the onset of neurological symptoms. miR-206, along with other suggested miRNA, have since been a key focus in multiple following studies [105,115,116] as clinical biomarkers, especially as miRNA may be too downstream and/or specific to be considered potential therapeutic target.

In a similar vein, Russell et al. (2018) investigated the regulation of select members of miRNA biogenesis pathways in pre-symptomatic, symptomatic, and end-stage SOD1^G93A^ mice, compared to the total miRNA levels at any different stage of disease progression [117]. They used muscle, brain, and spinal cord tissues, and measured miRNA and transcript levels using quantitative PCR. Interestingly, there was an increase in several genes involved in miRNA biogenesis as the disease progressed as well as in the total ratio between the examined miRNA and total RNA ratio, although uniquely in the tibialis anterior (TA) muscle. This relationship was not elicited in both the soleus and neural tissue, which implies that a dysregulation in the miRNA:tRNA ratio in the TA muscle from SOD1^G93A^ mice may be possibly reflected in pathological increase in miRNA biogenesis machinery.

Furthermore, the recent work from Weiner’s laboratory implicated the role of miR-155, a glial-enriched miRNA, in its upregulation in the spinal cords of end-stage ALS model (SOD1^G93A^) mice, a result similar to those previously discussed in the 2015 Butovsky study [118]. As Russell et al. (2018) have assessed different stages of ALS progression in mice, it is a confirmation of observations shown in Hoye’s study [119]. Thus, by inhibiting miR-155 significantly, disease progression in SOD1^G93A^ mice show the potential to be delayed [107,118], ultimately reinforcing the relationship that SOD1 and miRNAs share. However, it must be cautioned that despite enthusiasm for this glial miRNA therapeutic approach, it has been hypothesized that MN miRNA changes in ALS may be equally or more important in MN loss and pathology, defining disease onset and ultimate fatality [120]. Nevertheless, motor neurons represent only 4% of the mouse spinal cord volume [121], and thus probing miRNA changes would remain challenging. These studies together indicate possible alterations in the ratio between miRNA/total RNA that have a bearing on the levels of reference non-coding RNAs, and may therefore potentially compromise the accuracy of commonly used miRNA normalization strategies.

Finally, mutational analysis has also demonstrated a potential link between pathogenic SOD1 mutations and RNA metabolism dysfunction through altered stress granule dynamics [122], thereby fortifying not only the role of SOD1 in RNA metabolism, but the overall significance of TDP-43, FUS, and SOD-1 in association with ALS. Despite this understanding of such RNA-modifiers, how some of these proteins functionally associate with uncharacterized proteins and between one another remains unclear, a critical obstacle in developing novel treatment targets. It is valuable to consider that the heterogeneity of ALS may thus be a result of different affected proteins, all of which can then cascade into the same dysfunction of RNA and miRNA regulation.

## 4. Defining Interrelationships and the Plausibility of a Two-Way Interaction between RNA-Modifying Proteins and miRNAs

Thus far, miRNAs and RNA-modifying proteins including RBPs represent two key interconnected regulators of transcript degradation. We have discussed the effects of TDP-43, FUS, and SOD1 in miRNA production, how miRNA dysregulation is inevitable in lieu of protein dysfunction, and the suggested miRNAs involved. However, it must also be noted that the examined literature for each protein in relation to miRNA dysregulation has often led to inconclusive evidence about specific miRNAs in ALS, instead depicting a wide spectrum of certain affected miRNAs, despite no evident connection between them; notably, research has been specific to mouse models or fALS patients depicting exclusive genetic mutation, whether it be in TARDBP, FUS, or SOD1. This is significant as presently, there may exist the possibility that affected miRNAs may only be present amongst certain phenotypic presentations of ALS, and thereby explaining the lack of convergence of miRNAs, beyond the need for further accurate study and analysis [123]. Again, one must also consider the possibility of miRNAs being yet to be further classed into smaller groups or ‘families’, whether they may be common across cell types or hold equivalents between neurons, glia, and more, and may therefore explain such heterogenous presence of specifically dysregulated miRNAs in ALS and other NDs.

However, to truly illustrate the interaction that exists between TDP-43, FUS, SOD1, and miRNA, a discussion of the self-propagating, prion-like behavior of the RNA-modifying proteins is warranted. Apart from TDP-43 and FUS, more than 40 additional RNA-binding proteins have been characterized from the human proteome that contain predicted prion-like domains [124], with multiple being associated with ALS including HNRNP A2/B1, HNRNP A1, EWSR1, TAF15, ATXN2, MATR3, and TIA1 [5,125]. Moreover, a recent study has investigated approximately 180 RBPs that display specific co- and post-transcriptional interactions with distinct pre-miRNAs [126], which only consolidates the extensive explorative potential of RBP- and miRNA-foci in future research.

Autoregulatory properties of both TDP-43 and FUS have also been delineated in the past decade, with the impact of miRNA dysregulation having extended toward both associated miRNA genes and their respective RNA-binding proteins via triggered stress responses, in addition to the shared aggregate-inducing capabilities amongst all three proteins [127,128,129,130,131]. These properties suggest significance in the role of such proteinopathies in continuing the pathogenetic death of MNs in ALS, and thereby inspire more careful consideration of these prion-like and RNA-disruptive behaviors amongst such pathologically prevalent and similar proteins [11].

### 4.1. Autoregulation in Propelling a ‘Doomed’ RBP and miRNA Relationship

RBPs have been proven to regulate miRNA biogenesis through miRNA 3′ end modification, the conversion between pri-miRNAs into pre-miRNA, and subsequent transcriptions [132], and therefore warrant a discussion on existing regulatory mechanisms of RBP function. Several miRNAs have also been shown to have cell- and tissue-specific regulatory patterns [133], where some of such regulatory events take place at the transcriptional level of the pri-miRNA, and many of which involve the same group of transcription factors, like E2F1 for proto-oncogene, c-Myc, that regulate both gene expression and miRNAs [132,134].

Considerable evidence points to deficiencies in FUS and TDP-43 nuclear function in contributing to neurodegeneration, and it is likely that other RBPs converge functionally. Recently, it has been shown that a novel FUS autoregulatory mechanism exists in which high levels of FUS mRNA negatively feeds back onto FUS pre-mRNA, resulting in nonsense mediated decay. Certain ALS mutants, however, escape this self-regulation, allowing their protein levels to accumulate, worsening ALS pathology [135]. Surprisingly, TDP-43 also binds to its pre-mRNA and promotes alternative splicing, resulting in mRNA degradation via nonsense-mediated mRNA decay [136], which provides a feedback loop that controls the homeostasis of FUS and TDP-43 levels, as is disrupted in mutant FUS- and TDP-43 ALS phenotypes [26]. Whether miRNA dysregulation is decided to occur upstream or downstream, this essential dysfunction in RBP autoregulation may therefore be significant in the development of such pathology in ALS patients, particularly as the abnormal cytoplasmic accumulation of mutant protein/s in ALS neurons becomes uncontrolled [26,137].

Notably, however, although there is now stronger detail regarding TDP-43 autoregulation, the trigger for this ‘doomed’ cycle of dysregulation has yet to be elucidated by research, with multiple points in the cycle all valid entry points for such predicted dysfunction.

Regardless, further considering the implications of the roles of endocytosis, autophagy, and other clearance mechanisms of RBPs [3,138,139,140], one can therefore begin to appreciate a more holistic image involving RBPs and miRNA dysregulation (Figure 1), which may assist this visualization of an intrinsic two-way interaction between these key molecules. Due to similar pathological misfolding and co-aggregations [100] involving SOD1, it would be interesting to investigate the possible existence of similar autoregulation or a clear feedback loop for SOD1, especially as miRNA dysregulation has been so strongly elicited in G93A SOD1 mouse models [107,114,117,118,119].

Finally, an excess of miRNA binding sites at positions where RBPs also bind further cements the existence of this two-way relationship, as is dysregulated upon disease development. Experimental data show that the RBPs and miRNAs can either compete for analogous target sites in the 3′ UTRs of target mRNA strands, or that RBP binding may cause the miRNA binding site to become more accessible to the RISC complex. Moreover, both miRNAs and RBPs can use similar enzymes for degradation of target transcripts in similar sub-cellular compartments, which suggests miRNA-RBP interactions extend into transcript decay, beyond sole miRNA biogenesis and maturation regulation [132]. Although the role of RBPs is unambiguous in ALS and other neurodegenerative diseases, this two-way interaction between miRNA and RBPs remains poorly understood in relation to protein aggregation, RNA metabolism dysregulation, interference with cellular processes, and disease development. Further studies aimed at identifying specific, disease-relevant miRNA-RBP interactions should be performed to assist our understanding of the role of RBP autoregulation and clearance in ALS, and inevitably, of miRNA itself in ALS.

### 4.2. On Stress Response and Prion-Like Similarities: for Better or for Worse?

Beyond autoregulation, we now must investigate the responses of RNA-modifying proteins between one another in light of systemic stressors, particularly in its similarity to prion misfolding, aggregation, and seeding, and thus the essentiality of their dysfunction in translating to our increasing focus on miRNA dysfunction. ALS pathology is known to begin from either single or multifocal sites before spreading through the neuroaxis in a spatiotemporal manner [141]. Several RBPs display strong linkage to ALS as these proteins share common structural and functional properties including prion-like or low-complexity (LC) domains, which play a vital role in mediating the prion-like protein response as known in several NDs [5]. ALS-linked mutations in these domains of RBPs associate with augmented aggregation or propensity to fibrillate, cytoplasmic mis-localization, and dysregulation of stress granule dynamics, which implies that these LC domains have a significant role in the development of the biochemically modified protein inclusions in ALS pathogenesis. It is plausible that this propensity to aggregate is almost a ‘faulty’ response to stress, given the role that stress granules (SGs) have in RNA metabolism, as they regulate protein translation during cellular stress [142,143]. The relationship between SGs and neurodegenerative disorders is well described in the literature, with many protein components of RNA granules being recruited to modulate formation of SGs, which include ATXN2, TDP-43, FUS, survival of motor neuron (SMN), and fragile X mental retardation protein (FMRP) [142,143]. However, it should be emphasized that not all ALS-associated RBPs have a shared LC domain; known exceptions are MATR3 and ATXN2, further suggesting that alternative pathogenic mechanisms may underlie ALS pathogenesis [5].

It is also noteworthy to consider that, like TDP-43 and FUS, SOD1 also displays these prion-like features of aggregation and misfolding, but also in seeding, thus prompting our suggestion of investigating non-RBP SOD1 with a similar mindset to our approach to dysfunctional RBPs in ALS pathogenesis. Human wild-type SOD1 (HuWtSOD1) has been particularly shown to propagate intercellularly via exosome-dependent and independent mechanisms [90], as is similar to its intracellular propagation in SOD1 mice [87]. In relation to SOD1 misfolding in sALS, where SOD1 mutation is excluded, recent studies suggest that non-native conformers of SOD1 may also participate in a unified pathological mechanism that is shared amongst different forms of the disease [144], which further suggests the lack of a central, homogenous cause behind more parochial approaches to ALS pathogenesis, especially when foregoing the significance of protein and miRNA dysfunction. Interestingly, another recent study supports this notion, noting that although no variation was found in extracellular vesicles between ALS patients and the controls, both MVs and EXOs were found to have increased in size, with both methods significantly enriched in toxic TDP-43, phospho-TDP-43, FUS, and SOD1 when compared to the controls [145].

Considering this, one should always remain wary of the likelihood of ALS being attributed to several etiologies, and thus a value exists in understanding this final “common ground” before the more downstream pathogenic effects that are known to characterize the disease, particularly as this appears to be the tight interrelationship between miRNAs and RNA-modifying proteins, especially RBPs TDP-43 and FUS. Overall, these common properties in TDP-43, FUS, and other RBPs may also warrant future investigations on the existence of any direct relationship with miRNA dysregulation, as this will also clarify the extent of the miRNA-RBP two-way relationship, while also providing newer, plausible targets for novel therapies.

#### 4.2.1. On Functional Aggregation: Considering Subnuclear Paraspeckles (SNPs)

As is the case with ALS, despite these heavy self-regulatory effects, RBPs are not immune to pathological changes. Both FUS and TDP-43 have been a focus for the past decade, with the latter strongly aligning with the pathology [146] in >95% ALS cases and in >90% of ALS-affected post-mortem spinal cords, while also being implicated in MN death [147]. This most likely culminates from protein misfolding and insolubility resulting in neurodegeneration, analogous to what is seen in Alzheimer’s and Parkinson’s diseases, where such aggregation only further implies the prion-like properties of TDP-43, FUS, and possibly all known RBPs in the context of ALS and NDs [13,148].

In recent years, these prion-like domains in RBPs have been a source of speculation as they prove to be essential for “functional aggregation” and building SNPs, regulators of gene expression via nuclear retention of RNA [149,150], lacking in vitro in human and mouse neurons [146], unless stimulated by proteosomal inhibition, viral infection, or other stresses [82]. Both TDP-43 and FUS have prion-like domains and evolutionarily conserved RNA recognition motifs (RRMs), which play a part in protein aggregation, nuclear-cytoplasmic shuttling, and their localization into SGs and P-bodies [151]. The latter would most likely synergize to promote protein aggregation, although, together, these interactions play a vital role in the regulation of RNA stability, processing, and in guiding cellular stress responses, with the role of TDP-43 accumulation recently highlighted in triggering dysfunction [26,152,153].

Moreover, the relationship that RBPs have with SNPs must then also be appreciated, as both losses and gains in RBP function may prove to be protective and potentially essential in their association with de novo SNP development. A recent study [82] into endogenous mutant FUS delineates a consistent impairment of paraspeckle-dependent miRNA production, which aligns with observed SNP hyper-assemblies in the spinal motor neurons of both sporadic and familial ALS-FUS patients. With SNPs recently identified in enhancing global miRNA biogenesis, particularly via post-transcriptional pri-miRNA processing, the Microprocessor [154], and regulating mitochondrial function [155], their association with disease severity is unsurprising through the lens of miRNA dysregulation and protein dysfunction, at least for mutant FUS-indicated ALS patients. Loss of TDP-43 has also been shown to augment SNP assembly in cultured cells, with downregulation of core proteins involved in miRNA maturation also promoting assembly [146]. Not only does this evidence cement the complexity of a strong miRNA- and RBP-based involvement in ALS pathogenesis, but SNPs may also prove to be a protective mechanism upon miRNA dysregulation, although their roles are yet to be specifically defined. Consequently, this may confer the intended protectiveness of the prion-like domains of RBPs, especially if this protective function of SNPs may prove to be effective, and potentially become exploited as a future therapeutic target.

#### 4.2.2. Further Studies on SOD1 and RNA-Binding Proteins Interrelationships and Their Implications for Future Research

Clinically indistinguishable ALS can also be caused by genetic mutations of SOD1, TDP43, or FUS, and can occur in the absence of known mutation as sporadic disease [156]. In their first study in 2012, Pokrishevsky’s team showed that the accumulation of pathological TDP-43 or FUS coexists with misfolded HuWtSOD1 in patient motor neurons, which could trigger the misfolding in cultured cells. It also demonstrated that cytosolic mis-localization of FUS or TDP-43 in vitro and in vivo in ALS patients may kindle wtSOD1 misfolding in non-SOD1 fALS and sALS, and how indirect induction of SOD1 misfolding is possible, especially as the lack of immunohistochemical compartmental co-localization of said proteins still propagated SOD1 misfolding beyond its site of inception. Interestingly, in a recent second study, Pokrishevsky and colleagues [129] further demonstrated that TDP-43 or FUS-induced misfolded HuWtSOD1 allowed its propagation from cell-to-cell via conditioned media (Figure 4), while also showing that HuWtSOD1 can act a substrate for propagated misfolding, without accompaniment by transmission of pathological TDP-43 or FUS. Pokrishevsky et al. [129] therefore suggests the high possibility of prion-like intercellular transmissibility and induction of endogenous misfolded HuWtSOD1 in recipient cells, which implies a more profound significance of a network, RNA-protein-centric, network-wide approach to ALS pathogenesis. As the first of its kind in its studies of intercellular transmissibility between MNs, further investigations should be conducted to confirm such transmissibility, as they have been similarly transpired between pathological MNs and microglia in ALS mouse models [147].

This connection between these RBPs and SOD1 is highly plausible, especially as various TDP-43 and FUS forms are known to associate with mitochondrial impairment [157,158], which would lead to the generation of free radicals and finally, SOD1 misfolding due to protein oxidation. This is in addition to other associated mechanisms contributing to misfolding such as proteasomal or autophagic degradation, both of which are also used by pathological TDP-43 and FUS [140,159] for clearance, despite reducing the efficient clearance of misfolded SOD1 [160].

These implications argue for the equal propensity for these pathological proteins to seed, intercellularly transmitting mutant or misfolded wild-type SOD1 through the release of naked aggregates by the dying cells, which, in turn, are taken up by macro-pinocyotosis and trigger seeded aggregation [90]. Alternatively, this can also occur through the release of disease-associated exosomes containing intraluminal and surface-associated misfolded SOD1, which may be taken up by recipient cells through direct membrane fusion [161].

However, another investigation showed the converse of this situation, in which administration of mutant SOD1 induced TDP-43 aggregation in the target cell population [162]. Similar observations have also been recorded in human patients, mouse models, and cell lines, where in this case, mutant SOD1 co-precipitated with TDP-43 aggregates from SOD1 mouse spinal cord and from a human ALS patient [6]. The apparent cooperativity between SOD1 and TDP-43 in pathological aggregates offers an enticing explanation of how TDP-43 can aggregate despite the absence of mutations in its own sequence. Mutations that were once thought benign may gain new pathological significance in ALS with further research in this direction, which consequently emphasizes the value of understanding multiple protein dysfunction involving RBPs. Whether dysfunction proves to be due to SOD1 triggering RBP aggregates of TDP-43 and/or FUS or vice versa, or with miRNA dysfunction therefore having led to such, or both, there is a deep, under-researched interrelationship between these proteins that may shed even greater light onto the pathogenesis of ALS.

### 4.3. Connecting microRNA into a Complex Proteomic Picture

Thus far, a complex perspective has emerged between the RBPs and SOD1 in ALS. In defining the intricacies of the functional dependencies and influences between these RNA-modifying proteins themselves including their prion-like capabilities, we can finally attempt to clarify the intrinsic two-way relationship between these proteins and miRNA. From our previous discussions on miRNA biogenesis and TDP-43, FUS, and SOD1 (see Section 3), it can be concluded that RNA-modifying protein activity can directly interact with miRNA production such as through Dicer. However, miRNA effects on protein levels do remain indirect under our current understanding of RBP autoregulation and stress response, which emphasizes our need to define whether miRNA dysfunction is downstream or upstream to protein dysfunction, despite both their essentiality in ALS pathogenesis [163]. miRNA alterations in stress responses are hence still debated as to whether such are causative or consequential, with recent studies suggesting that stress-induced response complexes (SIRC) can form and increase endogenous miRNA targeting of nuclear RNAs [164], while RNA self-assemblies may also play a role in influencing significant dysfunction upon imbalance 118]. Mutations across TDP-43, FUS, and SOD1 have also long been known to trigger stress response pathways that ensue in global miRNA reductions [67], with associated deregulated pathways confirming multiple disease-related mechanisms including synaptic vesicle and cell cycle dysregulation before downstream MN degeneration, but these mechanisms fail to confer ALS specificity, or at least effectiveness in therapy [163]. Future miRNA-based therapies must therefore be investigated before this can be determined.

In a recent study by Brennan et al. [123], clusters of miRNA dysregulation were already found to be suggestive of possible functional convergence across NDs, and more interestingly, is the involvement of the prion disease pathway across all NDs including ALS without mutations to the otherwise causative PRNP gene. Notable examples include the association of miR-26a with ALS, Parkinson’s disease, and Alzheimer’s disease [123]; miR-26a particularly elicits an upregulation preclinically in human prion disease before downregulation to basal levels and further reduction during disease progression [165,166]. This, along with the overlap of prion-associated miRNAs with NDs [123], elicit the strong possibility of these prion-like properties of ALS-associated proteins to be critical in the propagation of pathogenesis, where further investigations should be conducted into representative changes in miRNA levels, should they be able to serve as biomarkers for disease progression, if not a potential avenue for treatment. Care must be taken in these studies as indicative from miR-26a studies, models may reflect a pre-clinical upregulation, for instance, before a drop to basal levels and thereafter, potentially continue to decrease. This initial upregulation and seeming basal level presence for an otherwise slowly-progressive disease like ALS may ultimately mask such changes if examinations are conducted at differing timeframes of disease study, and therefore may contribute to the presence of conflicting data.

As direct miRNA-to-protein interactions remain uncertain, it is also noteworthy that alterations in protein levels are instead attributed to a mix between pro-aggregation signaling and cellular mis-localization. Although most analyzed TDP-43 ALS-linked mutations accelerate aggregation in vitro [60], most FUS mutations trigger cytoplasmic accumulation of FUS or TDP-43 inclusion bodies, but not aggregation [10]. For TDP-43, the mutants Q331K and M337V, for instance, are aggressive in promoting aggregation in comparison to the wild type. In contrast, the FUS mutants H517Q, R521C, and R521H do not affect the aggregation kinetics of FUS [167], although the presence of the FUS mutations in the proline-tyrosine nuclear localization signal, PY-NLS, implies their possible role in disrupting cellular localization rather than its aggregation [168]. Together, these data support the concept that specific ALS-related mutations in TDP-43 manifest disease by promoting aggregation, whereas the ALS-related FUS mutations work by disruption of nuclear localization of FUS. These data also thus imply the complexity of the networks involved in ALS pathology, representing numerous dysfunctional reaction chains and sources of error. As a result, it is difficult to point at a single pathway or a mechanism by which all these RBPs functionally converge to cause ALS or neurodegenerative disease, leaving further questions as to whether the significance of miRNA influence on RBPs lies more upstream or downstream. Moreover, the direct effects of miRNA on stress granules or the prion-like domains of RBPs themselves remain relatively unknown, although further investigations, especially in relation to NDs and ALS, would be indispensable in understanding prevention strategies for this destructive aggregation phenomenon. Recently, some studies have uncovered the potential of miR-335 to promote stress granule formation in acute ischemic stroke [169], and more interestingly, the downregulation of miR-335-5p in contributing to MN mitochondrial dysfunction and apoptosis in ALS serum [170].

Regardless, since protein dyshomeostasis unifies several neurodegenerative diseases, and given the functional resemblance between RBPs, it is plausible to hypothesize that targeting ribonucleoprotein particle (RNP) assembly in human neurodegenerative diseases could be an effective strategy in the treatment of NDs. A better understanding of this process could shed much needed light on the mechanisms of regulation of this dynamic molecular machinery that governs specific cellular and physiological functions to maintain a non-disease state, and thereby elicit plausible future therapies, especially the potential for a common therapeutic strategy.

## 5. Discussion

Thus far, we have synthesized the importance of the three key misfolded proteins linked to ALS —TDP-43, SOD1, and FUS—whilst defining their relationships, interdependencies (Figure 1 and Figure 2), and current critical pitfalls in understanding in a RNA metabolism scope. As is evident, data indicate that no one protein is causative of pathogenesis in ALS, and we therefore must look toward approaching interactions with another common denominator to ascertain other possibilities for therapy, even if such prove to only be delaying disease progression.

### 5.1. Examining miRNA Regulation as a Potential Diagnostic Tool

Extensive downregulation of miRNAs has been discussed to be a common molecular determinant in different forms of ALS in humans, with pathogenic RBP mutations notably interfering with DICER and therefore, miRNA biogenesis, which then associates with abnormalities within the stress response system, as observed in ALS and other NDs. Whether miRNA dysregulation is upstream or downstream in pathogenesis is yet to be confirmed, however, the common ground of miRNA and RNA regulation, and RBP dysfunction, is ultimately undeniable across both sALS and fALS patients. In this context, Hornstein’s group [67] recently deduced global miRNA downregulation to be specific to affected motor neurons in the spinal cords of sALS patients, while also exploring a novel mechanism for modulating miRNA biogenesis under stress, involving stress granule formation and re-organization of Dicer and AGO2 protein interactions with their partners. They proposed cellular stress to affect pre-miRNA processing including ALS-causing genes with either overexpressed wild type or mutant forms of FUS, TDP-43, and SOD1 in NSC-34 cells, and elicited decreased DICER activity. Neuromuscular function may thus be improved by enhancing DICER activity via a small molecule such as enoxacin or quinoline analogues, as shown by two independent ALS mouse models. This therefore suggests the role of miRNA biogenesis downstream of the stress response, implying that Dicer and miRNAs affect neuronal integrity and could be possible therapeutic targets [67] (Figure 1 and Figure 2). However, it remains noteworthy to not disregard the possibility of miRNA dysregulation upstream in pathogenesis, as ALS remains a heterogeneous disease after all. At the very least, there is promise in using miRNAs as a diagnostic tool, given the extreme heterogeneity of sALS [80].

As a diagnostic tool, miRNAs were first investigated in 2012 by de Felice et al., where the team highlighted the potential for peripheral leukocyte profiling after successful comparison of miR-338-3p to healthy controls [171]. The same group have since expanded their analysis, confirming the involvement of miR-338-3p in ALS patient blood and neuromuscular junctions [172]. Raheja et al. also recently elicited a similar promise for serum miRNAs in correlation to clinical parameters for ALS, identifying four upregulated and one downregulated miRNAs against controls [116]. Multiple further investigations have also examined the differential expression of miRNAs across post-mortem tissue of ALS patients compared to the controls as well as in other tissue including muscle, cerebrospinal fluid, blood, and MN progenitors [115,163,173,174,175]. Furthermore, a recent study by Brennan et al. [123] revealed that miRNAs isolated from body fluids of neurodegenerative disease patients (including ALS patients) converge on similar functional pathways including the prion pathway and the ubiquitin proteasome system [123]. This further supports the use of miRNAs as biomarkers and indicates that miRNAs may be disease driving features across several NDs.

### 5.2. Examining miRNA Regulation as a Potential Avenue for Future Treatment

Novel treatment regimens may thus exploit the intricate two-way interaction between miRNA biogenesis and RBPs, especially as we can infer that miRNA dysregulation is likely to be the most critical upstream link, proving essential to the regulation of both (1) the RNA-binding proteins themselves (indirectly affecting their autoregulation), and (2) other gene expressions. Such genes include both STMN2 [176] and UBQLN2, the latter of which, when downregulated, is known to promote RBP aggregation through an impairment of degradation [177], further promoting an already vicious, self-perpetuating cycle. However, what initiates this cycle remains unknown, along with multiple specifics of the properties and roles of associated proteins and miRNA. For instance, a recent study has shown that toxicity in ALS does not develop through a single aggregate structure or aggregation process, given the significant differences between FUS and TDP-43 aggregation processes and subcellular localizations of each aggregate [178]. Critically, we must also emphasize that both TDP43 and FUS are not only RNA-binding proteins, but also nuclear proteins, while SOD1 is mainly located in the cytoplasm [87]. A clear understanding of this shuttling in conjunction with the miRNAs associated with this process will shed the much-needed light on underlying pathogenic mechanisms in ALS, and will thus enable the development of a new generation of therapeutic strategies including antisense oligonucleotide based strategies that can modulate this event.

Encouraging are the recent developments on the therapeutic front at Prosetta Biosciences, San Francisco, USA, where they have established cellular models recreating the mis-localization of TDP-43 from nucleus to cytoplasm and the stress-induced aggregation of the cytoplasmic mis-localized TDP-43. The team identified several Hitfinder compounds that can prevent aggregate formation in stress granules, while also discovering others that instead confer dramatic re-localization of TDP-43 back to the nucleus. Moreover, these results with cellular models were complemented by the ability of both classes of compounds to relieve the motor paralysis displayed by a TDP-43-based transgenic C. *elegans* animal model. SOD1, however, presents a more complex situation, being recently implicated in a protective role upon nuclear re-localization [87]. As a point of therapy, miRNAs targeting key proteins may truly prove possible, as studies involving artificial miRNAs through AAV vectors prove to effectively silence the SOD1 gene in macaques [103,179].

Further developments have also arisen, with a 2019 mouse study introducing viral-mediated antibody delivery to target the RRM1 degradation domain of TDP-43 inclusions [180]. Showing no adverse response and a decrease in TDP-43 aggregates, neuroinflammation and cognitive and motor decline, immunotherapy may hopefully be viable in humans. Similarly targeted therapies include the use of heat shock proteins, small inhibitors of TDP-43 aggregation, and nuclear import receptors, as described in a recent review [54]. Thus, these RNA-binding proteins hold immense potential as targets in ALS, as illustrated through such investigations.

Finally, regarding our discussions in understanding the close interrelationship between SOD1 and RBPs TDP-43 and FUS, this re-examined perspective of the non-RBP may help researchers better understand the seemingly causative heterogeneity exhibited in the common mutant SOD1 fALS patients and those non-mutant SOD1 ALS patients. This may therefore act as a consolation in devising treatment approaches for such a heterogeneous disease, particularly as treatment response may therefore effectively affect beyond that single target (e.g., SOD1), even if such a target proves to be downstream to central pathology, since ALS appears to be deeply integrated with multiple points of cyclical failure. In this respect, Liu et al. [181] have shown some promising observations through active immunization using the SOD1 exposed dimer interface (SEDI) peptide in SOD1^G37R^ transgenic mice, which resulted in reduced accumulation of misfolded SOD1 in the spinal cord, coupled with the increased survival by an average of 40 days. Thus, there appears to be a viable immune-therapy approach against misfolded SOD1 in fALS and sALS, which can be further expanded to other misfolded proteins.

### 5.3. Beyond Motor Neurons: Applying Connections between miRNA and RNA-Modifying Proteins to Holistically Understand Pathogenesis

Finally, it is also worthy to consider the relationship miRNA holds between the potentially self-propagating pathological RBPs (as discussed in Section 4.1) considering the existence of a similar self-propagating event between motor neurons and morphologically activated microglia. In ALS, microglia are long known to be linked to the slow degeneration of neurons [17,120,147], although the converse connection between neuronal degeneration and its furthering of microglial activation is rarely highlighted. This itself is suggestive of a similar positive feedback loop to RBPs that results in the secretion of neurotoxic molecules, which hence promotes the resultant neurodegeneration through self-perpetuation [120]. As potent regulators of gene expression through post-transcriptional fine tuning [20], miRNA has been suggested as a critical factor in microglia activation [182], and therefore, in further downstream understanding of ALS pathology (Figure 1). Further investigations into miRNA dysregulation may therefore have a more extensive significance in understanding diverse ND diseases including ALS, as the picture grows to be more complex, more network-based, and highly interrelated with RNA metabolic regulation and protein function.

## 6. Conclusions

Here we have reviewed common proteomic themes that are not only mechanistically important in ALS, but may also shed light on proteinopathies and their molecular basis that are involved across several neurodegenerative diseases. To date, no single entity or event results in slowly progressive neurodegenerative diseases, and it is believed that several processes encompassing environmental, epigenetic, and genetic events may define a disease phenotype. Given this, the next generation of treatments should focus on combinatorial therapies that are selective for several of these decisive events that define a disease phenotype. As discussed, investigating the role of microRNAs and RNA-binding proteins provides only a snapshot of the causative mechanisms in ALS pathogenesis. However, their significance in disease progression is greatly recognized, and its unified discussion was necessitated. Since in several NDs, specific nuclear or cytoplasmic protein accumulation forms the causative neuropathological picture, it will be important to identify and ideally classify microRNAs regulating the translation of these targets, while studying the quantitative effects on the proteome of these therapeutic miRNAs. Although still hampered by the heterogeneity of ALS, this insight into these crucial players will hopefully guide future research attempts into more effective diagnostic advancements and potential avenues for therapy.

## Figures and Tables

**Figure 1 ijms-21-03464-f001:**
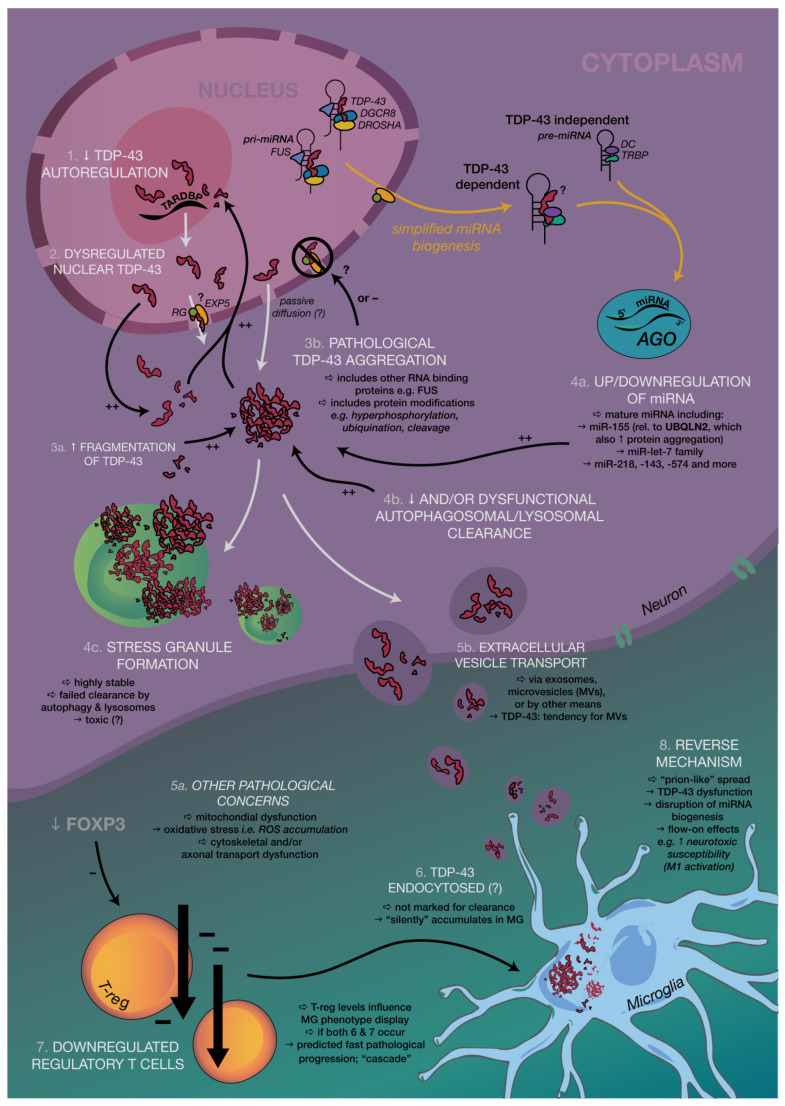
Simplified summary of key pathological mechanisms in a ‘network’ approach to Amyotrophic Lateral Sclerosis (ALS) involving transactive response DNA-binding protein (TDP-43), neurons, microglia, and regulatory T (T_reg_) cells. See Appendix A for further details.

**Figure 2 ijms-21-03464-f002:**
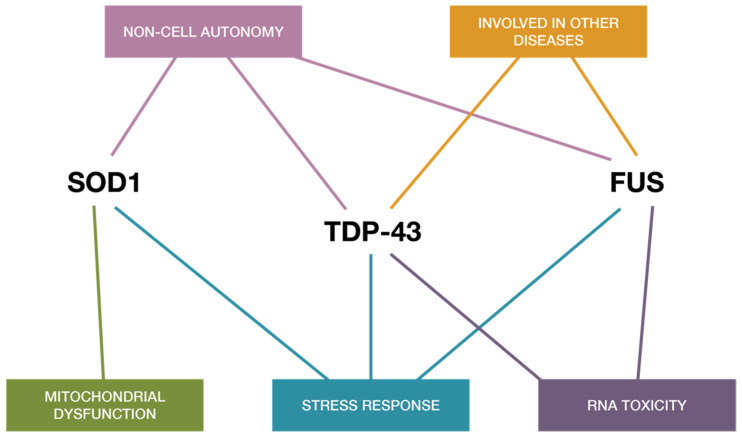
Common shared pathways in ALS and functional similarities shared between TDP43, SOD-1, and FUS. The three proteins appear to share mechanisms related to toxicity mechanisms and RNA toxicity, cellular stress response, and mitochondrial impairment and cell autonomy, implying functional synergies in ALS disease pathogenesis.

**Figure 3 ijms-21-03464-f003:**
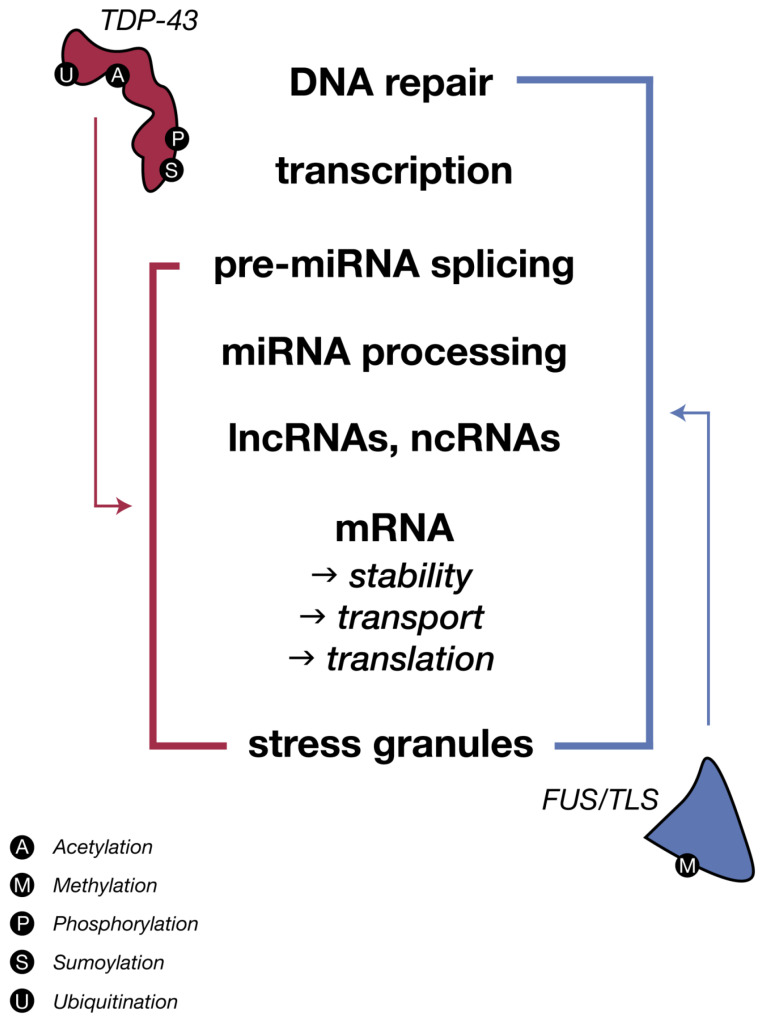
The two major ALS RNA-binding proteins, TDP-43 and FUS/TLS, share regulation of all aspects of RNA (metabolism, biogenesis and life cycle), despite variations in their biochemical processes. Schematic diagram depicts the sharing and distribution of molecular functional between these two major proteins. Figure 3: Idea adapted from Therrien M and Parker AJ. Worming forward: ALS toxicity mechanisms and genetic interactions in C. *elegans*. Frontiers in Genetics. 2014. https//doi.org/103389/fgene:2014-0085.

**Figure 4 ijms-21-03464-f004:**
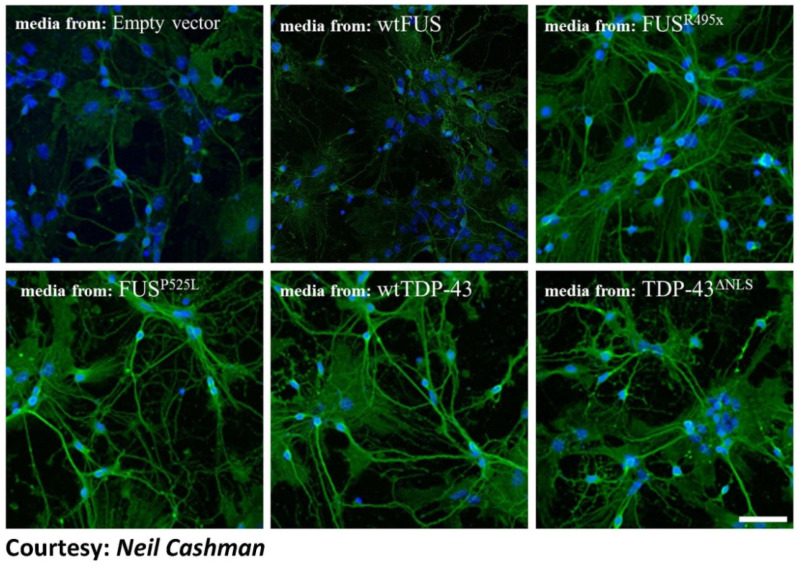
TDP-43 and FUS-induced misfolded HuWtSOD1 propagates from transfected cells to untreated spinal cord cultures. Primary spinal cord cultures containing neurons (including motor neurons) and astrocytes prepared from human HuWtSOD1 transgenic mice were incubated for 20 h with conditioned media from transfected HEK293 cells, and stained for misfolded SOD1 (green) using misfolded SOD1-specific antibody, 3H1, and counterstained using Hoechst 33342 (blue). The source of the media is indicated for each panel. Primary cultures incubated with conditioned media from FUSR495X, FUSP525L, wtTDP-43, and TDP- 43ΔNLS showed an increase in the presence of cytoplasmic misfolded SOD1 when compared to cells incubated with conditioned media from cells transfected with empty vector control and wtFUS. Scale bar: 75 μm. Figure 4 has been adapted from Grad, L.I., Pokrishevsky, E., Silverman, J.M., and Cashman, N.R. (2014). Exosome-dependent and independent mechanisms are involved in prion-like transmission of propagated Cu/Zn superoxide dismutase misfolding. *Prion* 8(5), 331–335, 2014. with the permission of Dr. Neil Cashman who is the corresponding author on the paper.

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
