# Peer review of "Connecting RNA-Modifying Similarities of TDP-43, FUS, and SOD1 with MicroRNA Dysregulation Amidst A Renewed Network Perspective of Amyotrophic Lateral Sclerosis Proteinopathy"

_ijms, 2020, doi:10.3390/ijms21103464_

Round 1

Reviewer 1 Report

In this manuscript the authors reviewed early and more recent literature studies regarding  the RNA biogenesis associated with altered function of RNA-binding proteins, including TDP43 and FUS, in the field of ALS pathogenesis. In particular, they focused on TDP43 and FUS functions in relation to microRNA regulation. Moreover, the authors described the contribute of SOD1 protein, that is not classified as RNA-binding protein, but has a relevant role in the microRNA metabolic dysfunction in motor neuron disease. The issue addressed in this manuscript is of interest in the field of ALS. The authors provided a critical discussion on this theme including good representative figures. However, from the reviewer’s point of view, major revisions, below reported, are needed to improve the manuscript.

As a whole, the manuscript is difficult to read, the paragraphs are too long and the same sentences are often repeated in different paragraphs (e.g. the sentence “…Drosha complexes with DCGR8 have been associated with most TDP-aggregates, which further suggests a more complex dynamic of protein pathologies [17,18]…” has been already written in the Introduction). The scope of the review is reported twice in the Introduction paragraph. The authors should merge the two sentences, modifying them, at the end of the Introduction.

The authors should carefully revise the paragraphs 2.1, 2.1.1, 2.1.2, 2.1.3, 2.1.4 focusing on the role of TDP43 associated with altered RNA metabolism in relation to microRNA: the paragraphs 2.1.1 and 2.1.2 could be merged becoming one short paragraph, as well as the paragraphs 2.1.2 and 2.1.3. Regarding 2.1.4 paragraph, the title is not clear. The authors should emphasize the link between TDP43 and altered microRNA biogenesis in ALS. In this paragraph the sentence “…As evidenced through multiple studies, TDP-43 affects several miRNA expression levels upon down-regulation, including induced pluripotent stem cell (iPSC)- derived neurons…” it is not clear whether the authors are referring to the studies conducted on iPSC-derived neurons or to the affected miRNA expression in iPSC-derived neurons.

In the paragraph 2.1.2 it should be added that TDP43 inclusions were also found in other cell types, e.g. skeletal muscle cells, apart from motor neurons and glial cells [Cykowski M.D. et al. Acta Neuropathol Commun 2018; Mori F. et al. Acta Neuropathol Commun 2019].

The authors mentioned Ago2 in the paragraph 2.1.2, but they should introduce it and explain its physiological function in the paragraph 2.1.1, in the miRNA biogenesis section.

The authors should revise the title of 2.2 and 2.2.1 paragraphs, because they are not clear enough. In particular, regarding 2.2 paragraph "Comparison to FUS-TLS" they should indicate which comparison are they referring to. Regarding 2.2.1 paragraph title, they should emphasize the link between FUS and altered microRNA biogenesis in ALS.

Also the title of 2.3.1 paragraph is not clear. The authors should emphasize the link between SOD1 and altered microRNA biogenesis in ALS.

Regarding paragraph 3, the authors should include a table to summarize the main interactions between RNA modifying proteins and microRNAs with the relative literature, for a better understanding by the readers.

The authors should revise the discussion and conclusion, they are too long and difficult to follow. They should focus on the principal issues to make more clear the key take-home messages.

The authors reported that Supplementary materials can be found at www.mdpi.com/xxx/s1, but the reviewer did not find supplementary material and the S1 file was not cited in the manuscript.

The list of abbreviations seems uncorrect.

Author Response

REVIEWER 1

As a whole, the manuscript is difficult to read, the paragraphs are too long and the same sentences are often repeated in different paragraphs (e.g. the sentence “…Drosha complexes with DCGR8 have been associated with most TDP-aggregates, which further suggests a more complex dynamic of protein pathologies [17,18]…” has been already written in the Introduction). The scope of the review is reported twice in the Introduction paragraph. The authors should merge the two sentences, modifying them, at the end of the Introduction.

This has been rectified, with further edits and the removal of the repeated sentence. Introduction has also been reshuffled to combine the two sentences, as recommended.

The authors should carefully revise the paragraphs 2.1, 2.1.1, 2.1.2, 2.1.3, 2.1.4 focusing on the role of TDP43 associated with altered RNA metabolism in relation to microRNA: the paragraphs 2.1.1 and 2.1.2 could be merged becoming one short paragraph, as well as the paragraphs 2.1.2 and 2.1.3. Regarding 2.1.4 paragraph, the title is not clear. The authors should emphasize the link between TDP43 and altered microRNA biogenesis in ALS. In this paragraph the sentence “…As evidenced through multiple studies, TDP-43 affects several miRNA expression levels upon down-regulation, including induced pluripotent stem cell (iPSC)- derived neurons…” it is not clear whether the authors are referring to the studies conducted on iPSC-derived neurons or to the affected miRNA expression in iPSC-derived neurons.

Review structure has been rectified to conform to recommendations as entailed, including the alteration of titles for clarity. The identified sentence has also been edited for clarity.

In the paragraph 2.1.2 it should be added that TDP43 inclusions were also found in other cell types, e.g. skeletal muscle cells, apart from motor neurons and glial cells [Cykowski M.D. et al. Acta Neuropathol Commun 2018; Mori F. et al. Acta Neuropathol Commun 2019].

This has been added in as requested.

The authors mentioned Ago2 in the paragraph 2.1.2, but they should introduce it and explain its physiological function in the paragraph 2.1.1, in the miRNA biogenesis section.

This has been rectified as recommended.

The authors should revise the title of 2.2 and 2.2.1 paragraphs, because they are not clear enough. In particular, regarding 2.2 paragraph "Comparison to FUS-TLS" they should indicate which comparison are they referring to. Regarding 2.2.1 paragraph title, they should emphasize the link between FUS and altered microRNA biogenesis in ALS.

These changes have been revised as recommended.

Also, the title of 2.3.1 paragraph is not clear. The authors should emphasize the link between SOD1 and altered microRNA biogenesis in ALS.

This has been modified to the recommendation.

Regarding paragraph 3, the authors should include a table to summarize the main interactions between RNA modifying proteins and microRNAs with the relative literature, for a better understanding by the readers.

The above recommendation is ideal, but due to further edits and explanation regarding the lack of convergence and knowledge of mentioned miRNAs, review authors believed it would be in the best interest to keep a clearer description of relevant miRNA instead.

The authors should revise the discussion and conclusion, they are too long and difficult to follow. They should focus on the principal issues to make clearer the key take-home messages.

Conclusion and discussion have both been cut down and revisited for clarity. If more revisions re length are required for further clarity, please let us know.

The authors reported that Supplementary materials can be found at www.mdpi.com/xxx/s1, but the reviewer did not find supplementary material and the S1 file was not cited in the manuscript.

This has since been removed.

The list of abbreviations seems incorrect.

This has been rectified.

Reviewer 2 Report

Pham J et al.  Int J Mol Sci

This review is on the changes in molecules involved in RNA metabolism and potentially related to ALS. Their focus is on TDP-43, FUS as RNA binding proteins (RBPs) and miRNAs as their regulator. They also described a possible interaction of SOD1 with RNAs.

They reported very details of the current knowledge about the above molecules, but the description is largely limited to the isolated findings, multiple lines of evidence indicate that RNA metabolism is not normal in ALS, insight into their relevance to the ALS pathogenesis is required, which is what we want to know most. For example, the role of WT TDP-43 in the RNA regulation is well describe, but previous reports have failed to specify what is wrong in the role of TDP-43 in ALS. We want to know how TDP-43 is abnormally regulated in ALS motor neurons (1. In Fig. 1). In addition, there is no description about how TDP-43 is fragmented (3a-1 in Fig. 1). Insight into how wild-type TDP-43 is abnormally processes as mutant TDP-43 is necessary to elucidate the pathogenesis underlying sporadic ALS. Although TDP-43 pathology observed in both sporadic ALS and TARDBP-linked ALS, the underlying mechanism may not be the same and TDP-43 pathology may be a consequence rather than a cause of ALS pathogenesis in the former. Implications of abnormal RNA metabolism in ALS without focusing on specific molecules cannot lead to development of therapy. Too strong tone of the description about the pathogenic role and therapeutic potential of RNA metabolism in general should be much softened.

Specific comments:

Title: As the description is mainly based on ALS model systems, original title may better be changed to e.g. “RNA-modifying similarities of TDP-43, FUS, and SOD1, molecules associated with Amyotrophic Lateral Sclerosis, and their functional dependencies in microRNA regulation

Line 33-34: Why only C9ORF72is indicated among the >30 ALS-linked genes so far identified? The proportion of C9ORF72-inked ALS varies among ethnicity, lowest in Asian countries (<1%) and highest in northern Europe (>30% of familial ALS and >15% of sporadic ALS).

Line 59: What do the authors intend to mean by “SOD1 and FUS positivity”? If they mean SOD1- or FUS-positive inclusion bodies, write as such. However, these proteins do not colocalize with TDP-43 in the inclusion bodies.

Line 76-77: Proteostasis of RBPs is frequently observed in ALS tissues, but this does not mean the ALS pathogenesis.

Line 93-94: SOD1 mutation is found only in < a few % of sporadic ALS patients, except in Northern Europe, where SOD1D90A mutation is prevalent in the population base. The proportion of ALS patients carrying TARDBP, FUS or ANG varies among different ethnicity.

Line 168-169: Studies by Yamashita et al. Nat Commun 3:1307, 2012 reported an insightful hypothesis as to the mislocalzation of TDP-43 in ALS motor neurons.

Line 294-5: Proteins and RNAs may non-specifically sequester to abnormal aggregates, In that case, RNAs may be sequestered in the SOD1 aggregates by chance. To discuss the pathogenic role of protein-RNA aggregation, discussion should be limited to specific binding.

Line 301: If the authors want to claim that WT-SOD1 plays some roles in ALS, proposed mechanisms underlying WT-SOD1 specifically acts toxic in ALS but not in other neurological diseases should be commented. In addition, the finding of SOD1 in association with RNA is not direct and an associated findings. Too strong tone of the role in RNA metabolism in other part is not appropriate.

Line 440-441: Although TDP-43 pathology is found in >95% of ALS cases, positivity of FUS pathology is much less, a few %.

Author Response

REVIEWER 2

Their focus is on TDP-43, FUS as RNA binding proteins (RBPs) and miRNAs as their regulator. They also described a possible interaction of SOD1 with RNAs.
They reported very details of the current knowledge about the above molecules, but the description is largely limited to the isolated findings, multiple lines of evidence indicate that RNA metabolism is not normal in ALS, insight into their relevance to the ALS pathogenesis is required, which is what we want to know most. For example, the role of WT TDP-43 in the RNA regulation is well describe, but previous reports have failed to specify what is wrong in the role of TDP-43 in ALS. We want to know how TDP-43 is abnormally regulated in ALS motor neurons (1. In Fig. 1). In addition, there is no description about how TDP-43 is fragmented (3a-1 in Fig. 1).

Details have been added in regards to this, or clarified if lacking in literature. In particular, TDP-43 fragmentation has now been briefly discussed as seen in section 3.1, as is appropriate.

Insight into how wild-type TDP-43 is abnormally processes as mutant TDP-43 is necessary to elucidate the pathogenesis underlying sporadic ALS. Although TDP-43 pathology observed in both sporadic ALS and TARDBP-linked ALS, the underlying mechanism may not be the same and TDP-43 pathology may be a consequence rather than a cause of ALS pathogenesis in the former.

This has been identified and discussed in Appendix A disclaimers regarding Figure 1, as well as modified suggestions within the paper.

Implications of abnormal RNA metabolism in ALS without focusing on specific molecules cannot lead to development of therapy. Too strong tone of the description about the pathogenic role and therapeutic potential of RNA metabolism in general should be much softened.

This has been rectified. If tone is still too strong, please let us know and we can seek to edit further as appropriate.

Specific comments:
Title: As the description is mainly based on ALS model systems, original title may better be changed to e.g. “RNA-modifying similarities of TDP-43, FUS, and SOD1, molecules associated with Amyotrophic Lateral Sclerosis, and their functional dependencies in microRNA regulation

Title has been modified, in conjunction to Reviewer 3’s similar recommendation, to: “Connecting RNA-modifying similarities of TDP-43, FUS, and SOD1 with microRNA dysregulation amidst a renewed network perspective of Amyotrophic Lateral Sclerosis proteinopathy.”

Line 33-34: Why only C9ORF72is indicated among the >30 ALS-linked genes so far identified? The proportion of C9ORF72-inked ALS varies among ethnicity, lowest in Asian countries (<1%) and highest in northern Europe (>30% of familial ALS and >15% of sporadic ALS).

The mentioning of the gene has been removed for better streamlined clarity.

Line 59: What do the authors intend to mean by “SOD1 and FUS positivity”? If they mean SOD1- or FUS-positive inclusion bodies, write as such. However, these proteins do not colocalize with TDP-43 in the inclusion bodies.

This has been clarified with references.
Line 76-77: Proteostasis of RBPs is frequently observed in ALS tissues, but this does not mean the ALS pathogenesis.

This has been rectified.

Line 93-94: SOD1 mutation is found only in < a few % of sporadic ALS patients, except in Northern Europe, where SOD1D90A mutation is prevalent in the population base. The proportion of ALS patients carrying TARDBP, FUS or ANG varies among different ethnicity.

This has also been rectified.

Line 168-169: Studies by Yamashita et al. Nat Commun 3:1307, 2012 reported an insightful hypothesis as to the mislocalization of TDP-43 in ALS motor neurons.

This has been added in as recommended, with surrounding background information reframed as needed.

Line 294-5: Proteins and RNAs may non-specifically sequester to abnormal aggregates, In that case, RNAs may be sequestered in the SOD1 aggregates by chance. To discuss the pathogenic role of protein-RNA aggregation, a discussion should be limited to the specific binding.

This has been addressed in discussing existing knowledge of SOD1-related aggregates, as well as the recognition of this pitfall in the existing literature.

Line 301: If the authors want to claim that WT-SOD1 plays some roles in ALS, proposed mechanisms underlying WT-SOD1 specifically acts toxic in ALS but not in other neurological diseases should be commented on. In addition, the finding of SOD1 in association with RNA is not direct and associated findings. The too strong tone of the role in RNA metabolism in other parts is not appropriate.

This has been amended as appropriate, with relevant references.

Line 440-441: Although TDP-43 pathology is found in >95% of ALS cases, the positivity of FUS pathology is much less, a few %.

This has been rectified.
Reviewer 3 Report

The review article by Jade et al has discussed a very interesting aspect of ALS proteinopathies involving three major factors namely TDP-43 , FUS and SOD1 and they scientifically argued the linkage between ALS proteinopathies in terms of RNA processing dysregulation and emerging roles of microRNA in the ALS pathomechanism. However, the manuscript requires significant revision before it can be accepted for publication in the IJMS. Please find the suggestions below:

  1. Although, the authors aimed to review the RNA modifying functional modulation of TDP-43, FUS and SOD1 depending on miRNA regulation, but the overall discussion ended up as the dysregulation of miRNA biogenesis as the outcome of specific ALS proteinopathy. I would suggest either change the word "dependencies" to match the discussion topic or add specific details indicating miRNA-regulated ALS proteinopathy onset.
  2. In the introduction part section 1, please remove C9ORF72 instead include info on target proteins (TDP-43 / FUS/ SOD1).
  3. Please cite original research articles wherever applicable instead of citing review articles, e.g. in the introduction, line 37, page 1, please include original citations along with Ref. 4.
  4. In Figure 1, include DNA repair in the illustration, and most importantly, please highlight common patho-mechanisms of FUS and SOD1 also, along with TDP-43.
  5.  in page 3, line 54, citations 6 and 7 are incorrect.
  6. in page 3, line 62-65, the statement does not match with figure 2 illustration, also in figure 2 the factors VAPB and C9ORF72 may divert the aim of the discussion, it's better to remove those factors.
  7. In page 3, line 75, please include more original citations along with Ref. 11 and 12.
  8. In page 4, line 90, citation 18 is incorrect, please rectify.
  9. In page 4, line 94, the factor ANG has appeared suddenly. Its better to replace it with SOD1 to keep the discussion focused.
  10. In page 5, line 130-132, the meaning of the sentence is unclear, please rewrite it appropriately.
  11. there are too many sub-sections under section 2, it will be easier for the readers if it can be arranged like one section for each of the three factors to discuss their roles in miRNA biogenesis.
  12. in page 5, line 159, please include DNA repair as one of the important function of TDP-43 along with appropriate original citation.
  13. In page 5, line 174, please change the case of "Ago2" to match that of mentioned in the later parts of the manuscript.
  14. It would be better if the sections 2.1.2 and 2.1.3 can be combined and discussed precisely.
  15. In the section 2.1.4, please include specific details of mentioned miRNAs in relation to ALS proteinopathies and neurodegeneration. Also the last paragraph of this section seems to be part of FUS section which has been misplaced. Please correct it.
  16. In page 7, line 236-239, please cite proper references. Line 241, correct spelling of "ubiquinated". Line 241-243, the statement is not correct as FUS also undergoes phosphoryaltion and sumoylation apart from its methylation, and Ref. 10 is quite older citation, please correct the statement along with the most recent citations. Line 245, please replace Ref. 59 with original article's citation.
  17. In section 2.2.1, the statement in line 247 is not true as TDP-43 and FUS overlapping pathologies have been reported by several ALS research groups. Please modify the statement with proper citation.
  18. In page 8, line 284, please replace the ref. 67 with more appropriate original references.
  19. In page 9, line 322, wrong laboratory citation, it should be Weiner's lab.
  20. In page 10, line 337, ref 83 is inappropriate. please change.
  21. In page 10, the first sentence of section 3.1 should be edited to clarify the meaning of the sentence, and also ref 84 is incorrect, please change. Line 361, please include the name of the particular transcription factors along with references.
  22. In page 13, line 503 include citation for TDP-43 also.
  23. Section 3.4 requires significant modifications to include specific mechanistic details linking miRNA to proteinopathies and oxidative stress. In line 562-563, there are studies on miR-335's involvement in stress granule assembly, please correct the sentence with proper info and ref.
  24. In discussion section, the authors suddenly started discussing about ALS drug therapy without preparing the ground of their mechanism in the previous sections, e.g. edaravone is a potent ROS scavenger, but the authors did not include oxidative stress in ALS in previous sections.
  25. In page 15, line 621, authors mentioned that SOD1 is always external to nucleus which is not true, please see the ref. PMID-26412972. Please correct the statement.
  26. In page 16, line 672-673, authors mentioned "our" but the ref. 25 is not from Saxena lab, please justify or replace the citation with appropriate one.
  27. References 137 and 138 are not cited in the text.
  28. The abbreviation section needs to be updated.
  29. The reference section requires editing.

Author Response

REVIEWER 3

Although, the authors aimed to review the RNA modifying functional modulation of TDP-43, FUS and SOD1 depending on miRNA regulation, but the overall discussion ended up as the dysregulation of miRNA biogenesis as the outcome of specific ALS proteinopathy. I would suggest either change the word "dependencies" to match the discussion topic or add specific details indicating miRNA-regulated ALS proteinopathy onset.

Title has been modified, in conjunction to Reviewer 2’s similar recommendation, to: “Connecting RNA-modifying similarities of TDP-43, FUS, and SOD1 with microRNA dysregulation amidst a renewed network perspective of Amyotrophic Lateral Sclerosis proteinopathy.”

In the introduction part section 1, please remove C9ORF72 instead include info on target proteins (TDP-43 / FUS/ SOD1).

This has been rectified.

Please cite original research articles wherever applicable instead of citing review articles, e.g. in the introduction, line 37, page 1, please include original citations along with Ref. 4.

This, along with future references to this issue, has been rectified.

In Figure 1, include DNA repair in the illustration, and most importantly, please highlight common patho-mechanisms of FUS and SOD1 also, along with TDP-43.

Appendix A has an additional paragraph regarding disclaimers in accordance to Figure 1, including the focus on a generalised pathological dysfunction, beyond the depiction of the normal roles of TDP-43. FUS and SOD1 similarities are also discussed, but due to their smaller prevalence amongst affected ALS individuals (with >95% of sALS patients exhibiting TDP-43 pathology), we have chosen to highlight TDP-43 as a key example to depict ALS proteinopathy in relation to factors beyond the motor neuron in the CNS, e.g. microglial involvement, as well as later immune system involvement.

in page 3, line 54, citations 6 and 7 are incorrect.

This has been rectified.

in page 3, line 62-65, the statement does not match with figure 2 illustration, also in figure 2 the factors VAPB and C9ORF72 may divert the aim of the discussion, it's better to remove those factors.

This has been altered as per recommendation.

In page 3, line 75, please include more original citations along with Ref. 11 and 12.

This has been rectified.

In page 4, line 90, citation 18 is incorrect, please rectify.

This has been rectified.

In page 4, line 94, the factor ANG has appeared suddenly. Its better to replace it with SOD1 to keep the discussion focused.

This has been rectified as recommended.

In page 5, line 130-132, the meaning of the sentence is unclear, please rewrite it appropriately.

This has been rectified.

there are too many sub-sections under section 2, it will be easier for the readers if it can be arranged like one section for each of the three factors to discuss their roles in miRNA biogenesis.

This has been rectified, as per Reviewer 1’s comments.

in page 5, line 159, please include DNA repair as one of the important function of TDP-43 along with appropriate original citation.

This has been rectified.

In page 5, line 174, please change the case of "Ago2" to match that of mentioned in the later parts of the manuscript.

This has been rectified.

It would be better if the sections 2.1.2 and 2.1.3 can be combined and discussed precisely.

This has been rectified as appropriate, as per Reviewer 1’s comments.

In the section 2.1.4, please include specific details of mentioned miRNAs in relation to ALS proteinopathies and neurodegeneration. Also the last paragraph of this section seems to be part of FUS section which has been misplaced. Please correct it.

Mentioned miRNA have extremely limited details, especially in terms of functional understanding, and this has now been included in the discussion. The final paragraph has also been moved as per request.

In page 7, line 236-239, please cite proper references. Line 241, correct spelling of "ubiquinated". Line 241-243, the statement is not correct as FUS also undergoes phosphoryaltion and sumoylation apart from its methylation, and Ref. 10 is quite older citation, please correct the statement along with the most recent citations. Line 245, please replace Ref. 59 with original article's citation.

References have been rectified, along with spelling corrected. Statement has also been modified with references. Mentioned references have been appropriately rectified.

In section 2.2.1, the statement in line 247 is not true as TDP-43 and FUS overlapping pathologies have been reported by several ALS research groups. Please modify the statement with proper citation.

This has been rectified with appropriate citation.

In page 8, line 284, please replace the ref. 67 with more appropriate original references.

This has been rectified.

In page 9, line 322, wrong laboratory citation, it should be Weiner's lab.

This has been rectified.

In page 10, line 337, ref 83 is inappropriate. please change.

This has been rectified.
In page 10, the first sentence of section 3.1 should be edited to clarify the meaning of the sentence, and also ref 84 is incorrect, please change. Line 361, please include the name of the particular transcription factors along with references.

These recommendations have been rectified as noticed.

In page 13, line 503 include citation for TDP-43 also.

This has been rectified.

Section 3.4 requires significant modifications to include specific mechanistic details linking miRNA to proteinopathies and oxidative stress. In line 562-563, there are studies on miR-335's involvement in stress granule assembly, please correct the sentence with proper info and ref.

Both points have been adopted into the relevant sections, including appropriate references.

In the discussion section, the authors suddenly started discussing ALS drug therapy without preparing the ground of their mechanism in the previous sections, e.g. edaravone is a potent ROS scavenger, but the authors did not include oxidative stress in ALS in previous sections.

This has been rectified, with the removal of the mentioned drugs and the brief addition regarding oxidative stress in ALS regardless.

On page 15, line 621, authors mentioned that SOD1 is always external to the nucleus which is not true, please see the ref. PMID-26412972. Please correct the statement.

This has been rectified as appropriate, with the addition of lines 725-728, along with rephrasing the original statement.

In page 16, line 672-673, the authors mentioned: "our" but the ref. 25 is not from Saxena lab, please justify or replace the citation with the appropriate one.

This has been rectified.

References 137 and 138 are not cited in the text.

References 137 and 138 are cited in Appendix A (reference numbers have since changed due to the addition of further references since the last submission).

The abbreviation section needs to be updated.

Rectified as per Reviewer 1.

The reference section requires editing.

This has been rectified.

Round 2

Reviewer 1 Report

The authors addressed almost all the relevant questions raised by the reviewer. Regarding the inclusion of a Table summarizing the main interactions between RNA modifying proteins and microRNAs, the authors’ reply is reasonable. The manuscript much improved; however, it needs few minor revisions listed below:

  • At the end of the Introduction the authors briefly report the topics, which will be considered in the review (…As a result, this review is framed around altered RNA processing, the significance of RNA-regulatory proteins and non-coding RNA molecules, and microRNAs in contributing to overall cellular and network dysfunction in ALS…). The reviewer suggests to list the topics in same order of their presentation in the manuscript.
  • A revision of the English is still needed.

Author Response

Reviewer 1

  • At the end of the Introduction, the authors briefly report the topics, which will be considered in the review (…As a result, this review is framed around altered RNA processing, the significance of RNA-regulatory proteins and non-coding RNA molecules, and microRNAs in contributing to overall cellular and network dysfunction in ALS…). The reviewer suggests listing the topics in the same order of their presentation in the manuscript.
    • The topics have been listed chronologically, as requested (lines 108-110)
  • A revision of English is still needed.
    • This has been attended to, and hopefully is now acceptable.

Reviewer 2 Report

The authors successfully improved the manuscript, making the manuscript much easy to read with accurate and prudent description. The readers now are able to understand the significant role of RNA metabolism in ALS pathogenesis.

L179-180: Authors in Ref. 57-61 described the role of C-terminal TDP-43 fragments in the cytoplasmic translocation but the authors in Ref. 62 hypothesized the importance of calpain-dependent cleavage in the formation of TDP-43 pathology in both sporadic ALS and TARDBP-associated ALS.

There are some typographical errors and mixing use of abbreviations and full terms, which may be corrected by the publisher.

Author Response

Reviewer 2
• L179-180: Authors in Ref. 57-61 described the role of C-terminal TDP-43 fragments in the cytoplasmic translocation but the authors in Ref. 62 hypothesized the importance of calpain-dependent cleavage in the formation of TDP-43 pathology in both sporadic ALS and TARDBP-associated ALS.
o Ref 62 as been moved to the mentioning of calpain-dependent cleavage as appropriate.
• There are some typographical errors and mixing the use of abbreviations and full terms, which may be corrected by the publisher.

Reviewer 3 Report

The review work by Pham et al has been significantly improved after the revision. The sections of the manuscript are now well organized and misleading references and info have been removed as well. The article is now suitable to be published in the IJMS after the minor revisions listed below:

  1. In page 8, line 284, please include the full form of SMA and add it to the abbreviation list.
  2. In page 7, line 246, please include references for FUS post-transcriptional modifications.
  3. Authors mentioned in the response 

    "On page 15, line 621, authors mentioned that SOD1 is always external to the nucleus which is not true, please see the ref. PMID-26412972. Please correct the statement.

    This has been rectified as appropriate, with the addition of lines 725-728, along with rephrasing the original statement."

    In the revised manuscript the lines (725-728) has been simply copy pasted from previous version of the manuscript instead of correcting it. Please rewrite the statement with inclusion of the appropriate original research references.

Author Response

Reviewer 3
1. In page 8, line 284, please include the full form of SMA and add it to the abbreviation list.
1. This has been amended as requested.
2. In page 7, line 246, please include references for FUS post-transcriptional modifications.
1. Appropriate references have been added (lines 246-247)
3. Authors mentioned in the response
"On page 15, line 621, authors mentioned that SOD1 is always external to the nucleus which is not true, please see the ref. PMID-26412972. Please correct the statement.
This has been rectified as appropriate, with the addition of lines 725-728, along with rephrasing the original statement."
In the revised manuscript the lines (725-728) have been simply copied and pasted from the previous version of the manuscript instead of correcting it. Please rewrite the statement with the inclusion of the appropriate original research references.
• Apologies – the original edit had quoted the incorrectly added lines (738-741), which detail the conditions of SOD1’s existence in nuclei. The revised lines (725-728) also now detail “while SOD1 is mainly located in the cytoplasm” instead.